# The Developmental Origins of Cancer: A Review of the Genes Expressed in Embryonic Cells with Implications for Tumorigenesis

**DOI:** 10.3390/genes14030604

**Published:** 2023-02-28

**Authors:** Savitha Balachandran, Aru Narendran

**Affiliations:** Departments of Pediatrics, Oncology, Biochemistry and Molecular Biology, Physiology and Pharmacology, Cumming School of Medicine, University of Calgary, 2500 University Dr. NW, Calgary, AB T2N 1N4, Canada

**Keywords:** developmental, tumorigenesis, embryonic, genetic, pluripotency

## Abstract

Tumorigenesis, which involves the uncontrolled proliferation and differentiation of cells, has been observed to imitate a variety of pathways vital to embryonic development, motivating cancer researchers to explore the genetic origins of these pathways. The pluripotency gene regulatory network is an established collection of genes that induces stemness in embryonic cells. Dysregulation in the expression genes of the pluripotency gene networks including *OCT4*, *SOX2*, *NANOG* and *REX1* have been implicated in tumor development, and have been observed to result in poorer patient outcomes. The p53 pathway is a highly important regulatory process in a multitude of cell types, including embryonic, and the tumor suppressor gene *TP53* is widely regarded as being one of the most important genes involved in tumorigenesis. Dysregulations in *TP53* expression, along with altered expression of developmentally originating p53 regulators such as *MDM2* and *MDM4* have been implicated in various cancers, leading to poorer prognosis. Epithelial–mesenchymal transition (EMT), the process allowing epithelial cells to undergo biochemical changes to mesenchymal phenotypes, also plays a vital role in the fate of both embryonic and neoplastic cells. Genes that regulate EMT such as *Twist1*, *SOX9* and *REX1* have been associated with an increased occurrence of EMT in cancer cells, leading to enhanced cell stemness, proliferation and metastasis. The class of RNA that does not encode for proteins, known as non-coding RNA, has been implicated in a variety of cellular processes and emerging research has shown that its dysregulation can lead to uncontrolled cell proliferation and differentiation. Genes that have been shown to play a role in this dysregulation include *PIWIL1*, *LIN28A* and *LIN28B*, and have been associated with poorer patient outcomes and more aggressive cancer subtypes. The identification of these developmentally regulated genes in tumorigenesis has proved to play an advantageous role in cancer diagnosis and prognosis, and has provided researchers with a multitude of new target mechanisms for novel chemotherapeutic research.

## 1. Introduction

The development, proliferation and migration of cancer cells have been observed mimicking critical pathways of embryonic development, providing a rationale for targeting these pathways in novel therapeutics’ development for cancer therapies. Over the last few years, various collections of developmentally vital genes, which remain relatively quiescent in normal tissues, have been identified in preclinical and clinical studies of cancer. This is a promising step forward for cancer therapy research, as these identified genes carry the potential to inform molecular pathways and regulatory factors that can be targeted to inhibit cancer progression. Normally, embryonic stem cells (ESCs), which are derived from the inner cell mass of the pre-implanted blastocyst-stage embryo, drive the proliferation of cells to facilitate embryonic development by their ability to be both self-renewing and pluripotent [1,2]. The diverse patterns of gene expression that promote these properties have also been observed in cancer cells, specifically in cancer stem cells (CSCs). The CSCs are the fraction of the tumor mass that are distinct in their capacity to promote self-renewal and tumor development. Such cells have also been shown to mediate cancer cell migration and invasion to facilitate subsequent tumor metastasis [3]. Thus, it has been proposed that the dysregulation of critical developmentally regulated genes may contribute to the initiation, development and metastasis of certain cancers. This review aims to provide an overview of the recent literature on a selection of genes that have been studied in relation to cancer, focusing primarily on the oncogenic mechanisms and cellular pathways they influence (Figure 1).

## 2. Pluripotency Gene Regulatory Network

It is well established that the pluripotent state of embryonic cells is controlled by a network of genes that encode for critical pluripotency transcription factors which promote an intricate network of regulatory events [3]. Genes commonly associated with this pluripotency network are: *OCT4* (also known as *POU51*), *SOX2*, *NANOG*, REX1, STAT3 and KLF4 [3]. Though determined to be the gatekeeper genes of the pluripotent state, recent studies have observed the specific processes by which disruptions in this network in progenitor cells may contribute to early tumor development. Disruptions in these circuits lead to the dysregulated expression of such transcription factors, inducing pluripotency-disrupting normal cell maturation and promoting the uncontrolled differentiation into cancer stem cells. 

The most commonly associated gene is *OCT4*, or octamer-binding transcription factor 4 [4]. Expressed in unfertilized oocytes, *OCT4* stimulates the initiation and preservation of pluripotency in the inner cell mass of blastocysts and epiblasts. It prevents the differentiation of blastomeres into extraembryonic trophectoderm cells, thus controlling pluripotency. Via epigenetic modifications such as DNA methylation and histone modifications, *OCT4* is silenced when differentiation is necessary. Embryonic stem cells contain hypomethylated *OCT3* gene loci which are rich with active histone marks to ensure pluripotency. In somatic cells, *OCT4* is silenced via DNA methylation in the promoter and enhancer regions, thus it is not expressed in normal adult cells. In cancer cells, *OCT4* has been established as a driver of neoplastic growth in various types of cancer, and has also been associated with poorer prognosis and shorter survival rates in clinical situations [4]. A recent study found that active histone methylation and acetylation marks, such as H3K4me3 and H3K9acS10p in the promoter region, were responsible for the overexpression of *OCT4* in breast cancer [4]. This study confirmed not only that neoplastic tissue contained a higher than normal expression of *OCT4*, but also demonstrated the functional importance of *OCT4* in tumor bulk. Silencing of *OCT4* lowered the rate of proliferation and increased apoptotic activity—indicating that *OCT4* plays a role in the anti-apoptotic characteristics of cancer cells. Furthermore, this study concluded that OCT4 plays a role in promoting cell migration, and enhances motility to support metastasis [4]. Enhanced expression of *OCT4* has also been observed contributing to resistance to chemotherapeutic drugs and tumor recurrence [5,6]. 

*SOX2* is another important gene in the regulatory network controlling pluripotency. *SOX2* is a part of the *SRY*-related gene family, and the expression of *SOX2* begins at the morula stage of the embryo [7]. It is expressed in both embryonic and extraembryonic cells during development and is observed in various tissues after birth, as opposed to *OCT4* which is silenced in adult tissue. *SOX2* appears to be essential for regeneration in adult tissue, though dysregulations of this gene have been observed in neoplastic cells [7]. Studies have uncovered the important role that *SOX2* has in cancer cell proliferation, anti-apoptotic properties and cell invasion in a variety of cancers [7]. Recently, it has been reported that *SOX2* is associated with rapid onset of metastasis and decreased survival rates [8]. Interestingly, *SOX2* has also been observed in association with changes in cancer cell metabolism such as enhanced oxidative phosphorylation, glycolysis and fatty acid metabolism, along with increased mitochondrial quantity, which is thought to contribute to the cells’ ability to metastasize [8]. 

*NANOG*, another gene that is associated with the pluripotency network, exhibits similar properties and clinical outcomes in cancer as the genes previously described [9]. During embryogenesis, it controls pluripotency in the epiblast and prevents the differentiation into the primitive endoderm. Elevated levels of *NANOG* have been observed in a variety of cancers, and are observed to be regulated by post translational modifications, specifically phosphorylation [9], Recent studies have observed distinct mechanisms by which high expressions of *NANOG* promotes tumorigenesis, including the ability for immune evasion [10]. 

*REX1* (reduced expression 1) is another gene involved in embryonic stem cells that promotes pluripotency [11]. Studies found that the hypermethylation of the promoter region of this gene contributed to reduced expression. This manifested clinically through increased cancer progression and advanced tumor stage [11].

Signal transducer and activator of transcription 3 (STAT3) has been shown to play a crucial role in the maintenance of pluripotency and somatic cell reprogramming through a number of mechanisms [12,13,14,15]. For example, in ESCs Leukemia Inhibitory factor (LIF) helps to maintain the undifferentiated state enabled by the activated STAT3 pathway [12]. However, Humphrey et al. have reported that in certain experimental conditions the maintenance of hES stemness may occur through STAT3 independent mechanisms [16].

The transcription factor KLF4 (Krüppel-like factor 4) has been implicated as functioning in the regulation of the gene expression programs involved in the maintenance of naïve pluripotency [17,18].

## 3. The p53 Pathway

The p53 pathway is a highly important physiological pathway that plays a vital role in genomic integrity and cell fate in embryonic, somatic and neoplastic cells [19,20]. The pathway is activated by stress signals, which initiates cell cycle arrest—allowing for the repair of damaged DNA; meanwhile, continued activation can result in senescence and/or apoptosis [19,20]. The tumor suppressor gene *TP53*, encoding for the p53 protein, is widely accepted as being one of the most important genes in tumor development, and is observed to be mutated or deleted in up to 50% of cancers [19,20]. The p53 protein, once activated, functions primarily as a transcription factor to regulate a large network of downstream processes that control cell fate [19,20]. A multitude of single nucleotide polymorphisms and nonsynonymous polymorphisms have been identified in the *TP53* gene (a vast majority in the DNA-binding segment region) which give rise to modified p53 function [21]. These mutations have the ability to render the protein incapable of regulating the downstream expression of genes, resulting in dysregulated occurrences of cell cycle arrest and apoptosis [21]. Mutations in this gene have also been observed to confer cancer-associated abilities to the mutated protein (referred to as “gain-of-function” activities) which can promote tumor development, metastases and increased resistance to anticancer therapies [21]. Studies have shown that the p53 protein promotes cell programmed death/apoptosis, allowing them to enter apoptosis in response to certain chemotherapeutics [22]. Specifically, certain mutations in p53 were observed to suppress the activation of the downstream p53 upregulated modulator of apoptosis (PUMA) protein, which is typically activated by chemotherapeutic agents to promote apoptosis of cancer cells. Other apoptosis-initiating downstream genes that are impacted by mutations in p53 include *BAX* (Bcl-2-associated X protein) and *NOXA* (Phorbol-12-myristate-13-acetate-induced protein 1), which are both essential for the apoptotic activity of the p53 pathway [22]. *MDM2* and *MDM4* are both genes that encode proteins that regulate the function of p53. *MDM2*, which is vital for embryonic development, controls low p53 levels in normal cell types by forming a negative feedback loop with the p53 protein (whereby increased p53 levels induce expression of *MDM2*, which leads to the degradation of p53). Overamplification of *MDM2*-encoded proteins has been observed in more than 17% of tumors [23]. Single nucleotide polymorphisms have been observed in the *MDM2* gene, correlating with tumor development and progression and an earlier onset of cancer occurrence [20,23]. The MDM4 protein functions alongside MDM2 to regulate p53, and its overexpression is also observed in cancers [23]. MDM4 binds to and inactivates p53 (as opposed to MDM2, which promotes its degradation) [23]. In addition, several of the single nucleotide polymorphisms identified in *MDM4* have been linked to various cancers and their aggressiveness [20,23].

## 4. Epithelial Mesenchymal Transition (EMT)

Epithelial-mesenchymal transition (EMT) is the process by which polarized epithelial cells of the basement membrane sustain various biochemical modifications to become phenotypically mesenchymal [24,25,26]. Mesenchymal cells are capable of cellular processes such as migration and invasion, and become more resistant to apoptosis, thus making it an important development in associated physiological processes. 

EMT is involved in three primary processes [24,25,26]. The first is during embryonic gastrulation and organ development, known as Type 1 EMT, in which the primitive epithelium (epiblast) transitions into primary mesenchyme, giving rise to the mesoderm, endoderm and mobile neural crest cells. The second process, or Type 2 EMT, is wound healing, tissue regeneration and/or organ fibrosis. Type 3 EMT is associated with neoplastic cells, specifically those with enhanced cancer progression and metastasis. Several studies have found that carcinoma cells adopt a mesenchymal phenotype by observing the expression of various mesenchymal markers (such as desmin, vimentin, α-SMA and FSP1), and it is these cells that enter into the invasion–metastasis cascade, establishing secondary colonies at distant sites in the body [24,25,26]. 

Various genes have been identified as promoters of EMT. Twist1, a transcription regulator during embryonic development, is an important driver of EMT and a key regulator of metastasis in cancer. A landmark study found that the suppression of Twist1 prevented the spreading of breast cancer cells to the lungs [27]. This has since led to the discovery of Twist1 as a key component in promoting metastasis in a number of other cancers [28]. 

Another important gene associated with EMT and cancer metastasis is *SOX9*, which specifically activates the HIPPO-YaP pathway [29]. The Hippo signaling pathway plays a central role in regulating cell proliferation and cell fate. This pathway is considered a tumor suppressive pathway which controls the expression of two proteins: YAP and TAZ—both of which are vital for tissue regeneration and organ development but which are also observed in tumorigenesis [29]. The YAP/TAZ proteins have been observed to promote cancer stem cell phenotypes, cell proliferation and metastasis and are often overexpressed in tumors [29]. Recent studies have found that the activation of this pathway can also promote EMT in carcinoma cells [30]. The silencing of *SOX9* resulted in reduced invasion and proliferation of carcinoma cells [30]. Increased expression of epithelial markers and the downregulated expression of mesenchymal markers have also been observed, indicating reduced occurrences of EMT [30]. Similar findings were also observed in esophageal squamous cell carcinoma, where the downregulation of both YAP and SOX9 resulted in the suppression of malignant phenotypes in tumors [30]. REX1, previously mentioned as a driver of pluripotency, has also been found to function as a promoter of EMT as a result of overexpression. In addition, studies have found that cells with overexpressed REX1 downregulated E-cadherin levels and activated the JAK2/STAT3 signaling pathway, which could play a role in neoplastic cell invasion and metastasis [31]. 

Another very important gene involved in the promotion of EMT is *SNAI1*, which encodes for the Snail1 zinc finger transcriptional repressor, a member of the Snail superfamily of transcription factors. The Snail superfamily of transcription factors, which are encoded by genes such as *SNAI1* (snail family transcriptional repressor 1) and *SNAI2* (snail family transcriptional repressor 2), is a group of proteins that is very important in EMT and needed for proper gastrulation and mesoderm formation in the developing embryo but in recent years has also been heavily implicated in tumorigenesis [32]. *SNAI1* induces EMT by binding to the epithelial-tumor suppressing gene CDH1, amongst other epithelial genes, repressing E-cadherin and claudins. It has been associated with elevated levels in a variety of aggressive cancers, and in chemotherapeutic resistance. For example, in an analysis of triple negative breast cancer cell models, the *SNAI1* repressor was associated with as many as 180 different genes, including genes that control cell differentiation and signaling. The suppression of *SNAI1* expression was observed to result in the altered regulation and expression of several hundred additional genes, resulting in a significant repression of cell motility in breast cancer cells [32]. Despite SNAI1 being a major transcription factor involved in the epithelial–mesenchymal transition, its silencing was not sufficient in the full reversion of the cells to a more epithelial phenotype, indicating the existence of an intermediary phenotype between epithelial and mesenchymal [32]. *SNAI1* has also been observed to be upregulated in hepatocellular carcinoma [33] and ovarian carcinoma [34] amongst several other cancer types, indicating its vital role in cancer progression, and its significance for prognostic and diagnostic tools in cancer research. 

SNAI2, occasionally referred to as *SLUG*, is another member of the Snail transcriptional superfamily that is associated with both embryogenesis and tumor development [35]. SNAI2 upregulation has been associated with altered expression EMT in breast cancer cell models and has been shown to contribute to enhanced incidence of metastasis and lower survival rates in aggressive cancer phenotypes [35]. Higher than normal *SNAI2* expression is associated with poorer prognosis, as observed in cases of pancreatic cancer [36], colorectal cancer [37] and ovarian cancer [38], indicating its important role in cancer development and migration. 

## 5. Non-Coding RNAs

Historically, genetic research on cancer cell physiology focused primarily on genes that encode proteins. However, recent discoveries of non-coding RNA (ncRNA) have shown that the class of RNA molecules that do not encode for proteins may also play a vital role in a variety of biological processes, including embryogenesis and tumorigenesis. Emerging research shows that ncRNAs play a vital role in the regulation, expression and communication of genes. Conversely, the dysregulation of the genes encoding RNA have been implicated in uncontrolled cell proliferation and differentiation [39,40]. 

An important gene that controls the ncRNA-mediated cell proliferation of both embryo and tumor development is the PIWI-like RNA-mediated gene silencing 1 (*PIWIL1*). *PIWIL1* is highly expressed in embryos at seven weeks of development, but is downregulated in subsequent weeks [41]. Expression of this gene leads to the production of piwi-interacting RNA (piRNA), a class of small ncRNAs that bind to PIWI proteins to form a piRNA/PIWI complex which contributes to silencing of gene expressions, genomic rearrangements and stem cell pluripotency maintenance [41]. In recent studies, PIWIL1 has been associated with enhanced cell proliferation and migration [42,43]. Clinical studies comparing PIWIL1 levels in malignant lung tissue to levels in normal lung tissue found that PIWIL1 was detected distinctively in tumor cells, whereas it was absent in normal tissue. The overexpression of *PIWIL1* contributes to increased proliferation rates, colony formation, cell migration and invasion [42,43]. Clinically, these effects manifested in a shorter time to relapse and lower overall survival rate. *PIWIL1* overexpression was also detected in breast cancer, and interestingly, it was found that the levels at which the gene was expressed can be used to classify different breast cancer subtypes [44]. Breast cancers that were negative for *PIWIL1* expression were more likely to be classified as Luminal A subtypes, whereas breast cancers with a high expression of *PIWIL1* were more often classified as Luminal B or Triple Negative subtypes [44]. Similar findings have been observed in a variety of other cancers including gastric, colorectal, ovarian, urothelial, nasopharyngeal and renal tumors [45,46,47,48,49]. 

Another important set of genes that drive the ncRNA-influenced development of embryonic cells and cancer cells includes *LIN28A* and *LIN28B* [49,50]. The expression of these genes promotes the downregulation of an important tumor suppressor, a microRNA (miRNA) known as *let-7* [49,50]. In embryonic stem cells, LIN28A post-transcriptionally regulates OCT4, an aforementioned driver of pluripotency, by binding to a coding region of its mRNA. Both LIN28A and LIN28B are found to be vital to embryonic survival [49,50]. *Let-7* represses a variety of oncogenes, and the loss of *let-7* function contributes to tumorigenesis [51]. Increased levels of LIN28A/B and low levels of *let-7* are found in a variety of human cancers, such as in liver, lung, ovarian, breast, colorectal and brain cancers [50,52,53,54]. 

The expression of *SNAI1*, a gene previously described in the EMT portion of this review and which is a part of the Snail superfamily of transcription factors, is also heavily implicated in and influenced by ncRNAs [55]. Several different miRNAs were observed binding to the 3′UTR region of the snail family transcriptional repressor 1 (*SNAI1*), resulting from the complementary binding sites between miRNA and these regions of snail. One prominent family of miRNAs which serve as key regulators of *SNAI1* is the miR-30 family. These miRNAs have been observed to target *SNAI1* mRNA in a variety of cancers, resulting in its inhibition. miR-22 has been observed to target *SNAI1* and inhibit EMT in tumor cells and their migration/invasion in a variety of cancers [55]. miR-153 is another miRNA that has been heavily implicated with *SNAI1*. Studies have reported that the downregulation of *SNAI1* by miR-153 results in the suppression of cancer phenotypes, allowing for miR-153 to serve as a potential prognostic marker [56]. Consequently, the downregulation of miR-153 has been associated with increased EMT and thus enhanced metastasis [56]. Furthermore, studies have found that miR-153 has the potential to enhance the effects of chemotherapeutics. The downregulation of miR-153 decreased pancreatic cancer cells’ sensitivity to chemotherapy drugs, while the transfection of miR-153 significantly inhibited tumor cell metastasis and increased apoptosis in chemotherapeutic resistant cancer cells [56]. This indicates that miR-153 and its influence on the Snail pathway may serve as a novel therapeutic target for chemotherapy resistance and as a preventative mechanism of reducing cancer metastasis [56] in a variety of cancers including laryngeal squamous carcinoma [57], malignant melanoma [58], lung cancer [59] and breast cancer [60]. 

Snail family transcriptional repressor 2 (*SNAI2*), another member of the Snail superfamily of transcription factors, is also heavily influenced by non-coding RNAs [55]. In recent studies, the epigenetic silencing of the microRNA miR-203 has been observed to upregulate *SNAI2* expression. This overexpression of *SNAI2* is associated with the increased invasiveness of malignant breast cancers, suggesting that in cases of aggressive and invasive cancers, miR-203 may be epigenetically downregulated or silenced [61]. On the contrary, the expression of the miRNA-181a has been observed to inhibit cell migration, invasion and metastasis by directly targeting the *SNAI2* pathway in cases of salivary adenoid cystic carcinoma [62], suggesting that this miRNA could be a potential therapeutic target in the prevention of metastasis in cases of aggressive cancers. In summary, non-coding RNA plays a highly important role in the regulation of genes implicated in both embryogenesis and tumorigenesis, and the emergence of research surrounding non-coding RNA is providing a plethora of new avenues in the discovery of novel cancer biomarkers and therapeutic targets.

## 6. Clinical Relevance

The identification of developmentally regulated genes in tumor development and metastasis plays a promising role in the clinical diagnosis and prognosis of cancers. Understanding the unique genetic makeup of the tumor allows for a more precise and individualized approach to diagnosis, and a more targeted approach to therapies. In recent advancements, research surrounding the specific genetic origins of cancers has allowed for the development of novel artificial intelligence diagnostic tools to guide medical professionals in their observations of cancer cases and diagnoses. These tools effectively supplement routine histopathological testing [63], providing a more molecularly-guided approach to tumor observation, and alleviating uncertainty in the diagnosis process, allowing for the generation of more effective treatment plans [63]. Furthermore, the advancement in knowledge surrounding developmentally regulated genes in tumorigenesis has opened the door to a multitude of potential chemotherapeutic targets for pharmaceutical research. In vivo administration of *let-7* miRNA, which was previously described to be downregulated in tumors through the control of developmentally regulated genes, was found to be effective against animal models of lung and breast cancer, suggesting that it could be the basis of a potential novel therapy [64]. Additionally, further research into the implications of EMT in cancer has identified significant metabolic pathways that are involved in the maintenance and promotion of EMT [64]. Thus, therapeutically targeting the metabolic utility between the epithelial and mesenchymal cell type states through the use of repurposed metabolic inhibitors shows promise in the reduction in metastasis incidence and improved patient outcomes [64]. As previously described, research surrounding the pluripotency gene network, a group of established inducers of pluripotency in embryonic cells and tumor cells, has found that its dysregulated expression has led to increased cellular resistance to chemotherapeutic agents. Recent studies have found that the inhibition of certain molecular targets involved in the pluripotency gene network has led to decreased chemoresistance in certain cancer types. For example, the inhibition of HDAC6, a histone deacetylase enzyme that is involved in the epigenetic regulation of pluripotency genes, has been proposed to increase therapeutic sensitivity in oral squamous cell carcinoma [65]. Knockout of *CHD1L*, a gene associated with tumor progression, has also been found to downregulate pluripotency factors, leading to decreased drug resistance, along with increased apoptosis of cancer cells and reduced cell proliferation, proving to be a promising target for novel therapeutics [66]. Ongoing research will continue to prove the important role of developmentally regulated genes in tumorigenesis, with the outlook of discovering and testing novel chemotherapeutic targets that will minimize the burden and improve outcomes for those living with cancer.

## Figures and Tables

**Figure 1 genes-14-00604-f001:**
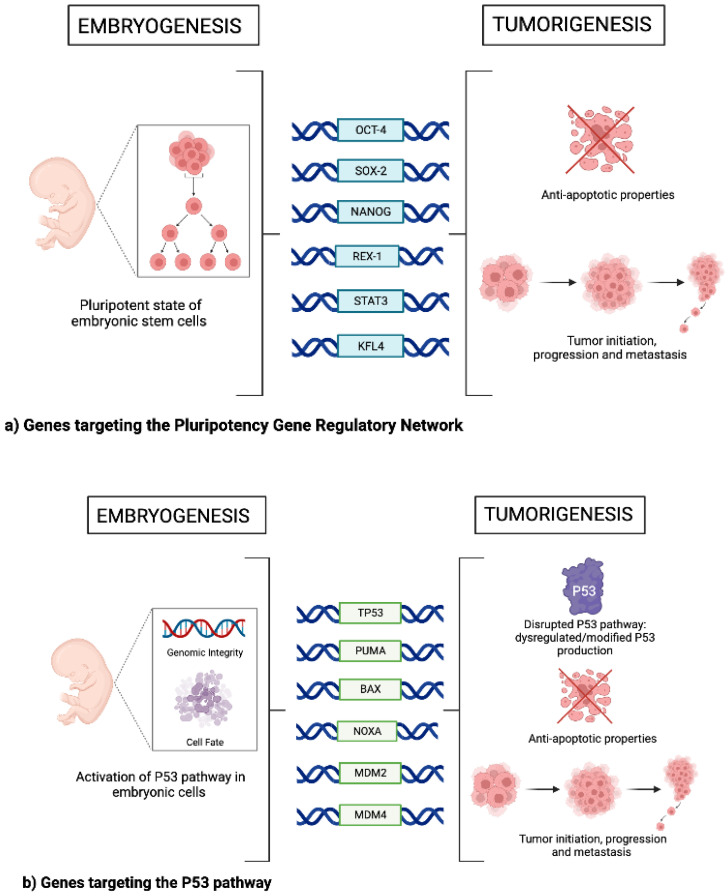
Key genes with regulatory implications for both embryogenesis and tumorigenesis. Functions and physiological outcomes of genes that target (**a**) the pluripotency gene regulatory network, (**b**) the p53 pathway, (**c**) that promote epithelial–mesenchymal transitioning and (**d**) the production of non-coding RNA.

## Data Availability

No new data were created or analyzed in this study. Data sharing is not applicable to this article.

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
