# Peer review of "The Developmental Origins of Cancer: A Review of the Genes Expressed in Embryonic Cells with Implications for Tumorigenesis"

_genes, 2023, doi:10.3390/genes14030604_

Round 1
Reviewer 1 Report
This review manuscript briefly summarized some major genetic determinants involved in both embryonic development and tumor development, with particular focuses on their oncogenic molecular mechanisms, cellular pathways or tumorigenic processes they influenced, and their clinical relevance. The manuscript is generally well organized and with good writing. Here I have two suggestions: (a) In the section of "Pluripotency Gene Regulatory Network" , the authors first listed the genes most commonly associated with this pluripotency network are OCT4, SOX2, NANOG, STAT3 and KLF4, then the authors discussed OCT4, SOX2, NANOG and REX-1 extensively, but STAT3 and KLF4 were not mentioned. The writing here need to be reconsidered for consistency. (b) It would be rather helpful to draw some schematics in a review paper to facilitate understanding and increase readability.
Author Response
Thank you for the careful and critical review of our manuscript. We agree with the revisions suggested and updated the manuscript accordingly, addressing all the requested changes. Revisions:- Information text has been added on STAT3 and KLF4 as requested by reviewer 1. (page 6 line 21-23, page 7 line 1-6)
- A schematic summary has been added as requested by reviewer 1 and 2 (Fig.1). (Page 3-5).

Reviewer 2 Report
This review, by Balachandran and Narendran, discusses the developmental origins of cancer in conjunction with a number of different contexts (stem cell gene regulatory networks and RNA-binding proteins etc.) that are normally overlooked. The review stands on an original combination and similar reviews are difficult to find. I think this review will be highly cited and could not find any points where I have to criticize for the text. One thing I would suggest is that the review does not have any figure or table; if they can make a representative figure as a blurb, it will help the readers to find the review intriguing.
Author Response

(The authors gave the same response as above.)
